# Improving Outcomes in Survivors of Sepsis—The Transition from Secondary to Primary Care, and the Role of Primary Care: A Narrative Review

**DOI:** 10.3390/jcm14082582

**Published:** 2025-04-09

**Authors:** Rosie Taylor, Sarah Vollam, Stuart R. McKechnie, Akshay Shah

**Affiliations:** 1Oxford Critical Care, Oxford University Hospitals NHS Foundation Trust, Oxford OX3 9DU, UK; rosietaylor5@gmail.com (R.T.); stuart.mckechnie@ouh.nhs.uk (S.R.M.); 2Nuffield Department of Clinical Neurosciences, University of Oxford, Oxford OX3 9DU, UK; sarah.vollam@ndcn.ox.ac.uk; 3NIHR Oxford Biomedical Research Centre, Clinical Informatics Research Office, Level 4, John Radcliffe Hospital, Oxford OX3 9DU, UK; 4Department of Anaesthesia, Hammersmith Hospital, Imperial College Healthcare NHS Trust, London W2 1NY, UK

**Keywords:** sepsis, critical illness, survivorship, quality of life, rehabilitation

## Abstract

Sepsis is defined as a life-threatening organ dysfunction caused by a dysregulated host response to infection. The number of patients with sepsis requiring critical care admission is increasing. At the same time, overall mortality from sepsis is declining. With increasing survival to hospital discharge, there are an increasing number of sepsis survivors whose care needs shift from the acute to chronic care settings. Recently, the phrase “post-sepsis syndrome” has emerged to encompass the myriad of complications in patients recovering from sepsis. The aim of this narrative review is to provide a contemporary summary of the available literature on post-sepsis care and highlight areas of ongoing research. There are many incentives for improving the quality of survivorship following sepsis, including individual health-related outcomes (e.g., increased survival, enhanced physical and psychological health) and wider socio-economic benefits (e.g., reduced economic burden on the healthcare systems, reduced physical and psychological burden on carers, ability for individuals (and carers) to return to workforce). Modifiable factors influencing long-term outcomes can be in-hospital or after discharge, when primary care physicians play a pivotal role. Despite national and international guidance being available, this area has been under-recognised historically, despite its profoundly negative impact on both patients and their families or caregivers. Contributing factors likely include the lack of a formally recognised “disease” or pathology, the presence of challenging-to-treat symptoms such as fatigue, weakness and cognitive impairment, and the prevailing assumption that ongoing rehabilitation merely requires time. Our review will focus on the following areas: screening for new cognitive and physical impairments; optimisation of pre-existing comorbidities; transition to primary care; and palliative care. Primary care physicians may have a crucial role to play in improving outcomes in sepsis survivors, and candidate interventions include education on common complications of post-sepsis syndrome.

## 1. Introduction

Sepsis is a life-threatening illness with acute and chronic implications for patients. In 2017, there were an estimated 48.9 million (95% Confidence Interval (CI): 38.9–62.9) cases globally, with 19.7% (95% CI: 18.2–21.4) resulting in sepsis-related death [1]. This equates to one person dying of sepsis every 2.8 s. There is also a significant health economic cost. In the USA, sepsis costs more than USD 24 billion annually [1]. In the UK, sepsis costs the National Health Service GBP 1.1 billion, alongside societal costs of up to GBP 10 billion per year [2]. The burden is disproportionately higher in low- and middle-income countries when compared with high-income countries, with mortality rates of around 50% versus 30–40% [3]. Priorities are also different. In high-income countries, these include lowering healthcare-associated infections and improving diagnostics and general management. Priorities in low- and middle- income countries include enhancing access to healthcare, improving sanitation, reducing malnutrition and overcrowding and promoting vaccination. Most published studies to date come from high-resource healthcare settings, and different approaches at a population level may be needed for low-resourced systems, for example broader prevention and control measures [3].

Survivors often experience poor long-term outcomes, and post-sepsis syndrome is now an active area of research. “Post-sepsis syndrome” is a description of the long-term complications arising after sepsis, including long-term physical, medical, cognitive and psychological complications. Management of post-sepsis syndrome is now a leading research priority [4].

There are many incentives for improving care and follow-up in sepsis survivors. In one study, 59% of hospitalisations involving severe sepsis were associated with worsened cognitive and/or physical function at one year post-hospitalisation [5]. Another multicentre cohort study shows that patients initially admitted with septic shock to the intensive care unit (ICU) were less likely to survive to one year compared with ICU patients without sepsis [6]. There is also a societal health/economic cost. Following admission for sepsis to the ICU, around 40% of survivors are readmitted to hospital in the first three months [7]. In one study [8], less than half of patients previously in permanent employment had returned to work at one year following ICU admission for septic shock. Between 5 and 15% [9] of sepsis survivors experience negative socioeconomic effects, such as inability to return to work at one year following hospital discharge; reduction in working hours or change in employment; and financial worries resulting in social isolation and psychological distress.

The aim of this narrative review is to provide a contemporary summary of the available literature on post-sepsis care and areas of ongoing research. We will focus on the following areas: screening for new cognitive and physical impairments; optimisation of pre-existing comorbidities; transition to primary care; and consideration for palliative care. Our emphasis is on critically ill patients who have required care in the ICU, but we acknowledge that many of these concepts may also apply to hospitalised, non-critically ill patients.

## 2. Methods

We searched relevant databases (PubMed, EMBASE) from 1 January 2000 until 1 January 2025. Subject headings entered were those that covered the population (adult and children survivors of critical illness), the exposure (sepsis, septic shocks) and outcomes focussing on the following areas: screening for new cognitive and physical impairments; optimisation of pre-existing comorbidities; transition to primary care; pharmacological treatments; and palliative care. Example search terms included “shock, septic”, “severe sepsis”, “sepsis”, “infections”, “all child”, “quality of life”, “post-sepsis syndrome” and “cognitive impairment”. Exclusion criteria included non-English studies, duplicate articles, conference abstracts and animal/pre-clinical studies. A range of publications were reviewed, including basic science, translational science, clinical trials, systematic reviews, narrative reviews, consensus guidelines and the most recently published clinical trials.

Where possible, studies were chosen specifically focussing on patients with severe sepsis and survivors of severe sepsis; however, where research is lacking in this area, studies from the general ICU population have also been referenced. The following sections of this narrative review will look at cognitive impairment following sepsis, optimisation of pre-existing comorbidities, the transition to, and role of, primary care in following sepsis, and palliative care.

## 3. Cognitive Impairment Following Sepsis

There is a large body of research showing that ICU survivors experience long-term cognitive impairment. This can range from memory loss and difficulty concentrating to anxiety, depression and post-traumatic stress disorder (PTSD). One study [5] in the USA, enrolling 1194 patients aged 65 and over, found that the prevalence of moderate to severe cognitive impairment at one year following hospitalisation was 10.6% higher (odds ratio (OR) 3.34, 95%CI: 1.53–7.25) among patients surviving sepsis, compared with no change in non-sepsis hospitalisation at one year following hospitalisation (OR 1.15, 95%CI: 0.99–2.15).

The aetiology behind long-term cognitive impairment following sepsis is multifactorial and includes disruption to the blood–brain barrier, neuroinflammation, neurotransmitter dysfunction and neuronal loss [10]. The degree of cognitive impairment is influenced by pre-existing comorbidities, severity of sepsis and quality of hospital care. There are potentially modifiable factors in a patient’s ICU stay that can be optimised to improve cognitive outcomes, and these include glycaemic control, minimising sedative medication and early mobilisation.

### 3.1. Glycaemic Control

Increased glycaemic variability, including both hyper- and hypoglycaemia, are associated with increased mortality in critically ill patients [11], including in patients with severe sepsis. The Surviving Sepsis Campaign [12] recommends a target of initiating insulin therapy for persistent hyperglycaemia > 180 mg/dL with a target range 140–180 mg/dL. However, there is limited evidence of the effect of glycaemic control in the ICU on long-term outcomes in sepsis survivors. A study in critically ill children, four years post-ICU admission and discharge, found that tight glycaemic control led to improved motor coordination and cognitive flexibility compared with a control group without tight glycaemic control [13]. Tight glycaemic control was also found to improve neurological outcomes in neurosurgical and neurological patients being cared for in the ICU [14]. However, a recent patient-level meta-analysis [15] of 38 eligible trials found no evidence of an effect on mortality of tight glycaemic control, including in the subgroup analysis of patients with sepsis. However, the confidence intervals were wide and may include clinically important benefits. Most studies define long-term impairment as follow-up at 12 months, and further studies are needed to investigate cognitive impairment at longer time points and whether certain subgroups of patients (e.g., those with pre-existing diabetes) may benefit from tight glycaemic control.

### 3.2. Early Mobilisation

A meta-analysis [16] comparing usual care to additional early active mobilisation in a heterogeneous population of ICU patients found no significant effect on days alive and out of hospital at 6 months following ICU admission. Some studies focussed specifically on the effect of early mobilisation on cognitive symptoms. One study, enrolling 60 general ICU patients, found that early mobilisation led to a reduction in psychiatric symptoms at three months, after adjustment for baseline characteristics [17], compared with the usual care control group (OR 0.27, adjusted *p* = 0.032). Psychiatric symptoms in this study were defined as at least one out of three of depression, anxiety or PTSD. Early mobilisation was defined as the rehabilitation level of sitting at the edge of the bed or higher within 72 h of ICU admission. In a single-centre RCT [18] of 200 general critically ill patients requiring mechanical ventilation, who were previously independent before ICU admission and were randomised to early mobilisation or usual care, the rate of cognitive impairment (measured using the Montreal Cognitive Assessment) at one year with early mobilisation was 24% compared with 43% of those with usual care (mean difference 19%; 95%CI 6 to 32%; *p* = 0·004). The Montreal Cognitive Assessment is a widely used screening test originally designed to detect mild cognitive impairment and early dementia. It has high sensitivity for detecting mild cognitive impairment. However, one preliminary validation study demonstrated only moderate concordance of this tool when compared with gold-standard tests [19]. Follow-up of a small RCT [20] from Australia followed 35 patients six months post hospital discharge and found that early physical rehabilitation for critically ill patients with sepsis resulted in an improvement in health-related quality of life in self-reported physical function using the SF-36 quality of life survey (mean (SD) 81.8 (22.2) vs. 60.0 (29.4), *p* = 0.04). Post-ICU rehabilitation guidelines [21] give a strong recommendation (high-quality evidence) for early mobilisation to be started within the first few days in the ICU, adapted to the patient’s resilience and general condition. It is unclear where differential effects are to be expected in patients being treated for sepsis.

### 3.3. Sedative Medication

Prolonged pharmacological sedation in ICU patients is associated with adverse outcomes [22]. High doses of benzodiazepines are associated with PTSD symptoms months after ICU discharge [23]. An RCT from the USA randomly allocated 187 patients to management with daily spontaneous awakening trials and spontaneous breathing trials, or to usual care. They found that a strategy using minimal sedation had no overall effect on quality of life outcomes compared to the control group; however, patients who received minimal sedation during ICU admission were less likely to report significant functional decline one year after ICU discharge compared with the control group [24]. Using minimal sedation may also lead to a reduction in length of hospital stay and ICU stay [25]. This has potential economic benefits and could lead to funding for resources post sepsis follow-up. A prospective study of patients admitted to the ICU with severe sepsis in Brazil [26] followed patients to one year post discharge. A cumulative haloperidol dose during ICU stay was inversely associated with anxiety (using the Beck Anxiety Inventory) after ICU discharge, and midazolam was not associated with anxiety. The Beck Anxiety Inventory is a 21-item commonly used self-report questionnaire designed to measure the severity of anxiety. However, its measurement properties and validity in post-sepsis survivors are unclear. The number of participants included in this study was also small (*n* = 33). An important consideration when interpreting data measuring cognitive impairment following severe sepsis is that baseline/pre-ICU admission cognitive status is seldom accounted for or adjusted for.

Guidelines [22] for multimodal rehabilitation following post-intensive care syndrome (PICS) have recently been developed which have the following strong recommendations: (i) firstly, interventions for delirium prophylaxis ought to include multimodal sensory, cognitive and emotional stimulation (mobilisation, purposeful stimulation and engagement, aids for orientation and contact to family members); and (ii) secondly, ICU diaries ought to be implemented for reducing the risks of symptoms of anxiety, depression and post-traumatic stress disorder in critically ill patients after discharge from the ICU.

## 4. Optimisation of Pre-Existing Comorbidities

It is difficult to detangle whether a worsening of clinical frailty, cognition and pre-existing comorbidities post sepsis are due to the critical illness sequelae or part of the pre-existing disease trajectory. It is most likely to be a combination of the two. Several studies have tried to separate contributions of the above to deterioration; however, it is challenging to obtain meaningful data of a patient’s pre-sepsis disease burden and predicted trajectory.

A study of 240 patients admitted to two ICUs in Scotland [27] showed that an increased number of comorbidities (measured using the Functional Comorbidity Index) was associated with worse health-related quality of life, as well as more patient-reported symptoms (appetite, fatigue, pain, joint stiffness, breathlessness), in all ICU survivors; however, this study was not focussed on patients with severe sepsis. In a retrospective cohort study of 1216 patients admitted to the ICU with sepsis in Singapore, mortality rates increased with Charlson Comorbidity Index [28]. Another study reported conflicting results. In a retrospective cohort study of 889 sepsis patients at one tertiary centre in the USA [29], pre-existing comorbidities were not associated with an elevated risk of physical impairment at hospital discharge (Risk Ratio (RR) 1.02, 95%CI 0.92–1.14). However, results were a binary of either no pre-existing comorbidity or pre-existing comorbidity.

Other studies have focussed on specific pre-existing conditions. McPeake et al. [30] investigated patient outcomes following critical illness by collating data from 3112 patients from the UK Biobank (a large prospective health resource). They demonstrated that patients with a history of clinical depression have worse long-term outcomes following critical care admission than those with no clinical diagnosis of depression (adjusted Hazard Ratio (HR) 1.49, 95%CI 1.14–1.96; *p* < 0.004). Targeted follow-up of patients with depression following critical care admission could help to improve outcomes. For example, one study following 126 patients across three hospitals found that a self-help manual reduced symptoms of depression [31] at eight weeks following ICU discharge (12% vs. 25%). Prescott et al. [7] found that readmissions after sepsis hospitalisation were common, and that around 42% were for conditions that could have potentially been prevented or treated early to prevent hospitalisation. Common readmission diagnoses included congestive heart failure, exacerbation of COPD and acute renal failure.

Studies show that there is an effect on socioeconomic deprivation and outcomes following critical care [32]. A national cohort study in Scotland looked at the impact of socioeconomic deprivation (as measured by postcode) on 30-day mortality in ICU patients with COVID-19. The study found that those living in areas with the greatest socioeconomic deprivation had a higher proportion of admissions to critical care, a higher adjusted 30-day mortality, a higher prevalence of comorbidity and a longer length of stay in critical care, compared to those living in areas of less socioeconomic deprivation. Further to this, ICUs in areas of socioeconomic deprivation spent more time above baseline bed capacity. Targeted follow-up after critical care admission to those patients with risk factors such as socioeconomic deprivation may guide the efficient use of resources. A population-based retrospective study of 23 years of data in Norway [33] looked at the effect of socioeconomic status on sepsis risk and mortality. Patients with low socioeconomic status had an increased risk of developing sepsis. Smoking and alcohol consumption explained 57% of the sepsis risk. Adding cardiovascular disease and chronic diseases to the model increased the explained proportion to 78% and 82%, respectively. In other words, approximately 80% of the increased risk in this cohort was explained by modifiable lifestyle and health-related factors that could be targets for prevention. Potential interventions can include patient (and carer) education at the point of discharge, reviewing and adjusting long-term medications, and telephone/digital follow-up for at-risk patients to monitor for new impairments.

Further research targeted at optimising specific pre-existing comorbidities prior to discharge, such as depression, heart failure and COPD, as well as their effect on patient outcomes following sepsis, is needed. This will help gain meaningful data that can be used to reduce long-term adverse outcomes following discharge.

## 5. Transition to Primary Care

As highlighted previously, around 42% of hospital readmissions following sepsis may be preventable [7]. Common reasons for readmission include new and/or recurrent sepsis, congestive heart failure and myocardial infarction. The most expensive reason for readmission in California was another episode of sepsis, with an estimated cost of USD 500 million/year between 2009 and 2011 [34]. A combination of early detection and patient education has the potential to decrease this, but it is unclear which patients may benefit. One study of over 90,000 adult sepsis survivors from ICUs in England identified the following risk factors for long-term mortality (with maximum follow-up to six years): age, male sex, one or more severe comorbidities, prehospitalisation dependency, non-surgical status, acute severity of illness, site of infection (e.g., respiratory) and organ dysfunction [35]. Another RCT developed a clinical prediction model to stratify sepsis survivors according to their risk of 30-day mortality and re-admission. High-risk features included older age; more comorbidities; recent hospitalisation; polypharmacy; and abnormal physiology and biochemistry (raised lactate, hypotension, lymphocytosis and tachypnoea) [36].

### 5.1. Ownership of Follow-Up

UK National Institute for Health and Care Excellence guidelines [34] for rehabilitation following critical care admission suggest that patients should have ongoing long-term monitoring; however, the guidelines are vague as to where overall ownership should lie for such follow-up. General practitioners (GPs) are well placed to care for such patients, as they may have a long-lasting relationship with the patients and have good knowledge of the most appropriate community care available for patient specific needs. A learning from the COVID-19 pandemic [37] was the need for improved communication between hospitals and primary care. In surveyed responses from 170 clinicians and GPs leading follow-up care after intensive care admission, 60% were unaware of the follow-up services provided by their local hospital. They reported that discharge documentation was often unclear and lacking key information, and there was an overall lack of clarity about who was responsible for referrals and follow-up.

The Surviving Sepsis Campaign [12] recommends the best practice of including information about the ICU stay, sepsis and related diagnoses, treatments and common impairments after sepsis in the written and verbal hospital discharge summary. However, this is not always possible due to difficulties in accessing secondary care records from primary care, limited understanding/knowledge of post-intensive care problems by ward staff writing discharge summaries and loss of potential key information due to multiple care transitions in the post-ICU period.

GPs have reported that they have so few patients out of their large patient populations who are discharged from intensive care that they are unable to take on the education/allocate resources for targeted follow-up [38]. There is scope for digital health solutions such as virtual clinics to manage patients post ICU stay [39]. These could include wearable devices to aid in the early detection of complications. Systems could be implemented to identify high-risk discharges and implement tailored follow-up; however, care must be taken when using wearables and other technologies to not widen healthcare inequalities.

### 5.2. Follow-Up Initiatives in Primary Care

Studies evaluating community-based interventions following severe sepsis admission have yielded mixed results. One multicentre RCT in Germany [40] followed up 291 sepsis survivors recruited in the ICU for one year. They had an initial follow-up in primary care of a one hour face-to-face session with the patient approximately one week after discharge from hospital. This was followed by monthly telephone contact for six months, and three-monthly telephone contact for the next six months. The use of primary-care-focussed team-based intervention compared to usual care was not associated with an improvement in the mental-health-related quality of life of patients 6 months after ICU discharge. There were some possible limitations to this trial—although the loss to follow-up was reasonable and the sample size significant, unblinded outcome assessments may have led to bias. Another multicentre RCT from Australia, which randomised 195 patients across 12 hospitals to an eight-week home physical rehabilitation programme versus standard care, did not observe any differences in the underlying rate of recovery over the 6-month follow-up [41]. A significant problem with this trial was patients lost to follow-up as they were unable to attend hospital outpatient clinics due to the distance to the hospital. This would be less of a limitation in a smaller country like the United Kingdom; however, it does suggest that home-based rehabilitation efforts may be more successful. A further nurse-led post-ICU follow-up programme [42] showed no improvement in QOL (and an increased cost compared to no intervention) one year after discharge from the ICU. The target trial size and interventions were delivered with high validity; however, the trial was generalised to all ICU patients.

Other studies have had more positive findings for community-based follow-up. One study found that a multicomponent sepsis transition and recovery programme reduced 30-day mortality and readmission outcomes after sepsis hospitalisation [43]. This programme occurred in North Carolina in 2019–2020 and was a nurse-led programme through telephone and electronic health record communication to facilitate best practice post sepsis. Care strategies included post-discharge medication review, evaluation for new symptoms, monitoring comorbidities and palliative care when appropriate. Some 691 patients were randomised and the intervention was associated with lower all-cause mortality and hospital readmission rates at 30 days post discharge compared with the standard (29% vs. 33%). The variability in outcomes of the above studies indicates that there is more work to be done to devise follow-up that is effective.

As alluded to in the previous section on co-morbidities, it is important to target specific patient groups in follow-up. One study [30] found that socioeconomic deprivation (measured by the Townsend Deprivation Index) increased long-term mortality (HR 1.31, 95% CI, 1.13–1.53 *p* < 0.001), and that social isolation is a risk factor for mortality following critical illness. These findings suggest ample opportunities for targeted post-critical illness care, namely targeted follow-up for patients with low socioeconomic status. Peer support groups could be a low-cost way to decrease social isolation following critical care admission. Support groups already exist for patients with severe sepsis, for example those run by the UK Sepsis Trust, who have 33 support groups in towns and cities across the UK [44]; however, more investment in this area could be beneficial to patient outcomes. The Surviving Sepsis campaign (2021) [12] recommends that best practice is to screen adults with sepsis and septic shock and their families for economic and social support (including housing, nutritional, financial and spiritual support), and to make referrals where available to meet these needs. This recommendation could be implemented during ICU stay or in the community after discharge.

### 5.3. Families of Survivors

It is important to consider the families of survivors too. NICE guidelines on life after intensive care also recognise the financial, emotional and health strain on families of survivors. Caregivers of ICU survivors in general have a poorer quality of life than caregivers of patients with Alzheimer’s disease [45]. Caregiver burden has been identified as an independent risk factor for both caregiver and care recipient mortality [46]. It is important to engage the families of survivors in care planning, education and training. This may happen in the primary care setting, due to existing relationships with the family. Such interventions will not only have a positive effect on the survivor, but also empower family carers, who find such a huge life adjustment and new caring responsibility “very overwhelming”.

A suggested roadmap for developing a post-ICU follow-up pathway is displayed in Figure 1. Virtual clinics and wearable technology can be used by healthcare professionals to reduce the burden on primary care physicians by asking patients to complete questionnaires to allow alerts and targeted follow up.

There is a need for a more unified approach with hospital, intensive care and primary care teams to achieve improvement in the management of patients following severe sepsis.

## 6. Palliative Care

There are circumstances in which a re-escalation of treatment in patients surviving sepsis, including admission to ICU, may not be appropriate. The Surviving Sepsis Campaign [12] recommends that best practice should include the discussion of goals of future care with patients and their family. There is little evidence to guide whether these discussions should occur early (within 72 h of admission) or later in the stay, but discussions within 72 h is the suggested recommendation, with the principles of palliative care integrated into the treatment plan where appropriate.

There is little in the literature about the best practice management of patients who have a further health decline following discharge from hospital following severe sepsis. Prescott et al. have suggested that in the 90 days after hospitalisation, if sepsis has occurred in the setting of long-standing comorbidity and declining health, the primary care team should discuss whether a transition to palliative focus is appropriate.

In a national retrospective cohort study in America between 2013 and 2014 [47], several factors have been identified after discharge that are associated with death within a year, including dyspnoea at rest, two or more hospitalisations in the past twelve months, living in an assisted living setting and overall poor health status. These patients may benefit from earlier in-hospital palliative care input, more frequent contact with primary care clinicians, advanced care planning and even hospice referral in some cases.

## 7. Conclusions

The number of sepsis survivors is rising, and targeted research is needed to better understand survivorship and to improve post-sepsis care. Sepsis survivors following critical illness experience lower health-related quality of life than population norms. Potentially modifiable factors include the use of sedative medications, glycaemic control and timing of mobilisation. Patients at increased risk of poor outcomes and increased healthcare resource use following sepsis include those with multiple pre-existing comorbidities, on polypharmacy or from lower socioeconomic backgrounds. This may serve as a useful starting point for enrichment strategies for any future sepsis survivor intervention trials. In addition, our review has also highlighted the heterogeneity in outcome reporting across studies, as also reported by others. Although core outcome sets exist for other conditions (e.g., acute respiratory failure survivors [48]), a bespoke set for sepsis survivorship is much needed.

### 7.1. Possible Areas for Improvement in Standard Care

There is a potentially growing role for primary care in post-sepsis management. Primary care teams are pivotal in both the recognition and management of the mental and physical health sequelae following critical care admission, as well as the large variety of complications in post-sepsis care. This is a very large burden of care for the primary health teams, for a relatively small patient cohort. In an evolving world of artificial intelligence, there is scope for data collation and analysis to target specific patients post-severe sepsis who would most benefit from specific follow-up in the community, as well as tools to detect early warning signs for patients following discharge.

To support this, we need to bridge the gap between intensive care admission and community care. This could be achieved by introducing a unified sepsis survivors handbook, held by the patient, which would provide a continuity of information across their care pathway. This could include documentation from ICU clinicians and subsequent follow-up in hospital and in the community, and perhaps signposting to resources for common side effects of post-sepsis syndrome.

### 7.2. Possible Areas for Future Research

There is a large body of evidence on outcomes for the general ICU population, but studies following survivors of severe sepsis years after discharge would be a beneficial addition to existing research. Studies looking at socioeconomic deprivation and the effect on outcomes in severe sepsis over a long time period would be valuable.

Further research on optimising specific pre-existing comorbidities prior to discharge, such as depression, heart failure and COPD, and their effect on patient outcomes following sepsis is needed.

Given the significant economic burden of sepsis, as outlined in this review, economic analysis on the cost-effectiveness of targeted follow-up and intervention in the management of sepsis survivors would be useful in progressing patient outcomes in a meaningful way at population level.

## Figures and Tables

**Figure 1 jcm-14-02582-f001:**
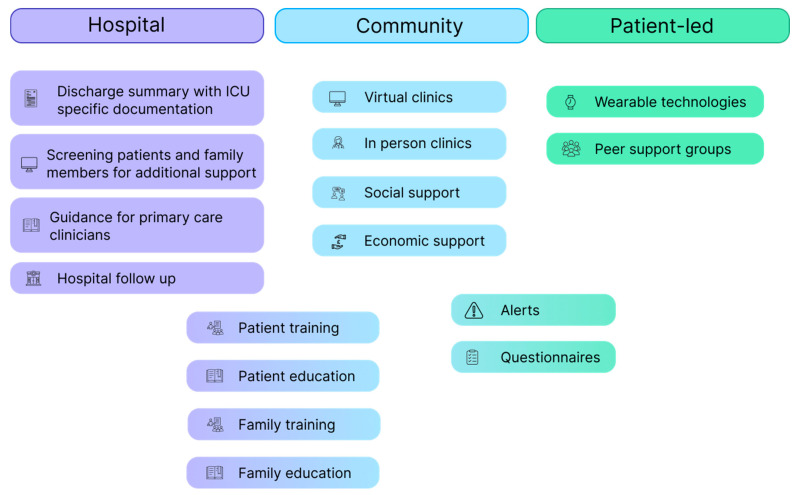
Follow-up roadmap.

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
