# Peer review of "Improving Outcomes in Survivors of Sepsis—The Transition from Secondary to Primary Care, and the Role of Primary Care: A Narrative Review"

_jcm, 2025, doi:10.3390/jcm14082582_

Round 1
Reviewer 1 Report
Comments and Suggestions for Authors
This narrative review provides a comprehensive and timely overview of post-sepsis care with a focus on the transition from secondary to primary care. The manuscript addresses an important clinical issue given the increasing survival rates after sepsis and the substantial burden of post-sepsis syndrome.
Major Strengths:
- Well-structured review covering key aspects of post-sepsis care
- Thorough discussion of modifiable factors during ICU stay
- Good integration of recent evidence, including studies up to 2024
- Valuable discussion of socioeconomic factors affecting outcomes
- Helpful roadmap figure for developing post-ICU follow-up pathways
Major Weaknesses:
- The methods section lacks detail on:
- Specific inclusion/exclusion criteria for studies
- Quality assessment methodology for included studies
- Number of studies screened/included
- PRISMA flow diagram would be helpful even for a narrative review
- Limited discussion of:
- International variation in post-sepsis care practices
- Cost-effectiveness analyses of different follow-up strategies
- Patient and family perspectives on care transitions
Specific Comments:
Introduction:
- At the beginning of the section, authors should provide more background on the global burden of sepsis. Please discuss doi: 10.3390/epidemiologia5030032
- Consider adding brief definition of "post-sepsis syndrome" earlier in introduction
Methods:
- Search strategy needs more detail
- Consider adding formal quality assessment of included studies
Results:
- Cognitive Impairment Section:
- Good coverage of evidence
- Consider adding more detail on screening tools and their validation
- Pre-existing Comorbidities Section:
- Well-structured analysis
- Could benefit from more discussion of specific intervention strategies
- Transition to Primary Care Section:
- Comprehensive coverage
- Consider expanding on implementation challenges
- Add more detail on specific tools for risk stratification
- Palliative Care Section:
- This section is relatively brief
- Consider expanding on timing and triggers for palliative care consultation
Figure:
- Useful addition but could be more detailed
- Consider adding specific timepoints for interventions
Author Response
"Please see the attachment"

Reviewer 2 Report
Comments and Suggestions for Authors
This narrative review by Taylor et al. provides a comprehensive overview of an increasingly important topic in critical care medicine - the management of sepsis survivors after hospital discharge. The authors have thoroughly examined the current literature regarding post-sepsis syndrome and various interventions that might improve long-term outcomes. They have structured their discussion around several key domains: screening for cognitive and physical impairments, optimisation of pre-existing comorbidities, transition to primary care, and consideration of palliative care. The review makes a compelling case for better integration between hospital and community care, highlighting the crucial role that primary care physicians can play in managing these complex patients.
Overall Assessment
In principle, this is a nice and timely review that addresses a significant gap in sepsis management. The authors have done a good job of synthesizing available literature from 2000 to 2025 and presenting it in a clinically relevant manner. Their proposed roadmap for post-ICU follow-up is particularly practical.
Points for Minor Improvement
The methodology section could benefit from more detail about the search strategy. While the authors mention databases and time period, it would be helpful to know the specific inclusion criteria and how they assessed the quality of studies they have included.
The review draws heavily from studies involving general ICU populations, not specifically sepsis survivors. It would be worthwhile to more clearly distinguish findings from sepsis-specific studies versus those from broader critical care populations, as the recovery trajectories may differ.
There is some inconsistency in how outcomes are reported across the cited studies. A brief discussion about the heterogeneity in outcome measures would help readers better interpret the evidence presented.
While the concept of a "unified sepsis survivors handbook" is intriguing, more concrete details about its potential content and implementation would strengthen this recommendation.
The review would benefit from more discussion about the cost-effectiveness of proposed interventions. Given the significant economic burden of sepsis mentioned in the introduction, economic analyses would be valuable for healthcare systems considering implementing these recommendations.
The applicability of findings to different healthcare settings could be addressed more explicitly. Most cited studies come from high-resource healthcare systems, and adaptation to other settings may require different approaches.
The proposed figure (Follow-up roadmap) is useful but could be enhanced with clearer indication of which components have stronger evidence behind them versus those that are more theoretical or expert opinion.
A more structured conclusion with specific recommendations for clinical practice and future research priorities would strengthen the practical impact of this review.
Overall, this is a valuable contribution to the literature that highlights an important aspect of sepsis care often overlooked in clinical practice and research. With minor revisions addressing the points above, it would be even more helpful for clinicians working to improve outcomes for sepsis survivors.
Author Response
"Please see the attachment"

Reviewer 3 Report
Comments and Suggestions for Authors
Dear authors,
Thank you for submitting your manuscript on sepsis outcomes and their relation with primary care. I found the manuscript interesting and topic-wise a good addition to existing literature on sepsis - the focus is far too often the acute phase and not what comes after.
The text is already well written and understandable to a first-time reader. There are a few points that should, however, be addressed before processing this further:
-) Title: I think the current title is a bit misleading or at least not clear enough. You focus on outcomes after sepsis has been survived by a patient, and the role of primary care in the following process. Please try to convey this message in the title. Also please add that it is a narrative review.
-) Methods: Thank you for already including a short Methods section. In my opinion, this should be extended and should at least include your full PICO and clear and transparent in- and exclusion criteria, the full search strategy, all databases searched, and all kinds of literature included, as well as the timeframe. Please don't just give examples but make your complete methods transparent here.
-) Main text: Maybe try to make the transition to chapter 3 (cognitive impairment...) a bit smoother, maybe with a transition paragraph before it? Otherwise you read about the methods and then suddenly cognitive impairment sort of pops up.
-) Main text: Please try to introduce subheadings to your primary care text as it is quite long and breaking it up a bit would enhance readability.
Author Response
"Please see the attachment"

Round 2
Reviewer 3 Report
Comments and Suggestions for Authors
Thank you for having addressed my comments.